# Multilayered Manufacturing Method for Microfluidic Systems Using Low-Cost, Resin-Based Three-Dimensional Printing

**DOI:** 10.3390/s25030694

**Published:** 2025-01-24

**Authors:** Victor Edi Manqueros-Avilés, Hesner Coto-Fuentes, Karla Victoria Guevara-Amatón, Francisco Valdés-Perezgasga, Julian Alonso-Chamarro

**Affiliations:** 1Instituto Tecnológico de la Laguna, Tecnológico Nacional de México, Cuauhtémoc y Revolución s/n, Torreón 27000, Coahuila, Mexico; hesnercf@lalaguna.tecnm.mx (H.C.-F.); kvguevaraa@lalaguna.tecnm.mx (K.V.G.-A.); fvaldesp@lalaguna.tecnm.mx (F.V.-P.); 2Group of Sensors and Biosensors, Department of Chemistry, Autonomous University of Barcelona, Edific Cn, 08193 Bellaterra, Barcelona, Spain; julian.alonso@uab.es

**Keywords:** microfluidics, resin-based 3D printing, fabrication

## Abstract

This work presents a multilamination method for fabricating microfluidic devices or analytical microsystems using commercial 3D printers and photocurable resins as primary components. The developed method was validated by fabricating devices for the colorimetric measurement of copper ions in aqueous solutions, achieving results comparable to traditional cyclic olefin copolymer (COC) systems. The microfluidic platforms demonstrated stability and functionality over a twelve-week testing period. Channels with minimum dimensions of 0.4 mm × 0.4 mm were fabricated, and the feasibility of using resin modules for optical applications was demonstrated. This study highlights the potential of combining 3D printing with multilamination procedures as a versatile alternative, offering flexibility through the selection of a variety of available resins and commercial printers, as well as the ease of design development. This method offers significant reductions in cost, time, and manufacturing complexity by eliminating the need for equipment such as CNC machines, presses, and ovens, which are typically required in other multilamination technologies like LTCC and COC.

## 1. Introduction

Microfluidics is a field of study that involves handling very small amounts of fluids. It has seen exponential growth, allowing for its application in a wide variety of scientific and technological fields, including biology, chemistry, medicine, and engineering. The precise manipulation of fluids using small channels has produced miniaturized systems that have been applied to medical diagnostics, disease detection, and chemical and biological analyses [1,2]. However, the selection of materials and fabrication techniques is a critical factor as it influences the efficiency, economic viability, and accessibility of these microsystems.

The first microfluidic devices were fabricated from silicon or glass using microelectronics techniques such as lithography and etching. These methods allowed for the production of highly precise, miniaturized, and high-quality devices, but at a high cost and with strict environmental requirements, such as the need for clean rooms [3,4].

Once microfluidic devices demonstrated their effectiveness, the development of new applications grew rapidly due to the use of novel materials and simpler, more cost-effective microfabrication methods. Among the prominent technologies is computer-controlled machining (CNC) using mechanical cutting and laser ablation, which is widely used on polymer substrates. These techniques enable the rapid and precise fabrication of microstructures, making them ideal for prototyping or the small-scale production of devices.

CNC micromachining is especially useful when working with thermoplastic substrates, as it provides a good balance among substrate cost, manufacturing speed, and surface quality. On the other hand, laser micromachining offers even greater precision. However, both technologies require equipment with costs ranging from medium to high, as well as specialized software.

Additionally, methods such as microembossing and microinjection remain essential for the mass production of microfluidic devices. These processes allow for the fabrication of devices with high-quality surfaces and high reproducibility, but they require considerably higher investments in specialized machinery to produce the necessary molds. This approach is primarily preferred for industrial production, as the initial investment is offset by spreading it over the high volumes of mass-produced devices [5,6].

Layer-based or multilamination manufacturing methods have recently gained popularity due to their ability to construct devices with complex geometries that cannot be produced cost-effectively using other methods. Multilamination involves creating a three-dimensional structure layer by layer using one or multiple materials, which may vary or remain the same.

This approach includes various micromachining techniques capable of defining different patterns and structures in each layer. The resulting three-dimensional structure can integrate diverse functionalities, such as microfluidic channels, electronic circuits, sensors, and other detection systems. Layer-by-layer construction allows for the creation of structures with material gradients or tunable mechanical or chemical properties, which is particularly valuable in biomedical and diagnostic applications. However, these approaches require specialized thermal compression equipment, such as adjustable pressure presses with heated plates, which can increase the initial investment in infrastructure.

Three-dimensional printing is an emerging technology in the fabrication of microfluidic devices. This technique encompasses methods such as stereolithography (SLA), fused deposition modeling (FDM), selective laser sintering (SLS), and multi-jet modeling (MJM). These techniques produce complex three-dimensional structures one layer at a time. Each method offers a unique combination of cost, processing speed, resolution, and design complexity. For instance, SLA and SLS provide the high resolution and precision necessary to produce complex microfluidic structures, albeit at a relatively higher cost. Conversely, FDM is a more economical option but has limitations in resolution and surface quality [5,6,7,8].

Although 3D printing is considered an additive layer-by-layer manufacturing technology, it is essential to distinguish it from other layer-based technologies, such as multilamination. In 3D printing, the final structure is created by segmenting the object into layers, each of which is formed and fused with the previous layer in a continuous printing process. In multilamination [9], the layers are fabricated separately and then aligned and processed using various methods (thermal compression, sintering, bonding, etc.) such that all layers merge into a single final three-dimensional object. Low-temperature co-fired ceramics (LTCC) and cyclic olefin copolymers (COC) are two materials commonly used to fabricate microfluidic devices using the multilamination approach. Both materials and their associated technologies present advantages and disadvantages depending on the application. Although development costs are significantly lower than those of other described methods, these methods remain beyond the reach of researchers with limited funding.

LTCC technology allows for the integration of microchannels, valves, sensing elements, and signal-processing circuits. This approach produces robust and precise microfluidic systems for biomedical, diagnostic, and chemical applications. However, LTCC-based microfluidic systems require special materials with adequate thermal compatibility, such as substrates, inks, and pastes, alongside expensive equipment like laser ablation machines, presses, and furnaces. These requirements can hinder the use of LTCC for low-cost prototyping.

COC, an amorphous high-performance copolymer, offers exceptional transparency and biocompatibility. By varying the copolymer composition, different glass transition temperatures can be achieved. The use of COC layers with varying glass transition temperatures allows for the creation of complex microfluidic structures using a simple multilamination process with thermal compression. These characteristics make COC an ideal material for applications requiring optical, visual, or biological interactions. Compared to LTCC, COC manufacturing processes are simpler and faster but still require CNC machines, precision presses with heated plates, or ultrasonic lamination machines [10,11].

Considering the high costs of the mentioned manufacturing technologies, the complexity of the production processes, the specialization of the programs used, and the limitations in the use of different materials and equipment by institutions with limited resources, the development of a low-cost device manufacturing methodology is proposed, combining the advantages of resin-based 3D printing (using commercial printers) and multilamination procedures. In the proposed method, machining and the need for thermal compression, which are characteristic of technologies such as LTCC and COC, are eliminated, making the proposed method a promising option for fabricating custom, low-cost microfluidic devices with small channels tailored to specific applications in analytical chemistry.

To fabricate the microfluidic devices that were used to validate the methodology, only an Epax resin printer (model X1) and a Sunlu UV curing chamber were used. The same resin served as the bonding agent for assembling the layers, eliminating the need for pressure and/or heat. This approach prioritizes simplicity over complexity and functionality over miniaturization.

The application of multilamination combined with resin-based 3D printing, as presented in this study, without the use of pressure and/or temperature, improves compatibility with materials sensitive to these variables, both inside and outside the devices. Using resin as a bonding agent also allows sensors and actuators to be fixed or embedded within the same device [12,13,14,15]. These features, along with the wide variety of resins available on the market, could extend applications to other fields such as medicine, the food industry, and environmental sciences.

Additionally, 3D printing does not require costly additional equipment, making this technology more accessible than LTCC- or COC-based manufacturing methods. This accessibility enables low-cost technologies to become available to laboratories worldwide, fostering innovation in the field of microfluidics [10,11,12]. The 3D printing techniques based on stereolithography (SLA) and digital light processing (DLP) used in this study are more accessible than other configurations, such as multi-jet printing (MJP) [16,17,18,19,20,21].

## 2. Materials and Methods

Three-dimensional printing was performed with an Epax printer model X1 (Epax, Morrisville, NC, USA) using 140 mm (5.5 inches) LCD SLA technology with 2K resolution (2560 × 1440 pixels) and 40 W power compatible with 405 nm photosensitive resins. ELEGOO ABS-like photopolymer resin (ELEGOO, Shenzhen, China) was used in four tints: clear, red, black, and green. A translucent resin supplied by ANYCUBIC (Shenzhen, China) was also used. The program Fusion 360 2.0.16490 (Autodesk, San Francisco, CA, USA) was employed to develop the 3D structures. The configuration and generation of the files provided to the 3D printer were produced using the ChiTuBox Basic 1.9.4 (ChiTuBox, Shenzhen, China) program for SLA/DLP/LCD printers. The data and dimensions were analyzed with OriginPro 2023B software (Northampton, MA, USA). The models were post-cured in a 405 nm curing box supplied by SUNLU (SUNLU, Irvine, CA, USA, EE. UU.), featuring a rotating plate and a 60 W lamp.

All reagents used throughout this work were of analytical grade. The stock copper solution for ICP (1000 ppm), disodium phosphate (Na_2_HPO_4_), monosodium phosphate (NaH_2_PO_4_), and nitric acid (HNO_3_) were supplied by Sigma-Aldrich (Darmstadt, Germany). 3-Hydroxy-4-nitroso-2,7-naphthalenedisulfonic acid disodium salt (NRS) was supplied by Fluka (Honeywell, NC, USA). Working copper (II) solutions were prepared via successive dilutions from a 10 ppm stock standard copper (II) solution. A phosphate buffer solution (adjusted to pH 6.6), a 1.1 mM solution of the NRS reagent, and a 0.01 M HNO_3_ solution were prepared using MilliQ water (Chemistry Laboratories of the Tecnologico de la Laguna, Torreón, Coah, Mexico).

Transmittance was measured using a UV–Vis spectrophotometer series AM1706003 instrument (Shanghai Metash Instruments, Shanghai, China). The microfluidic modules printed in resin were designed based on a lock and key configuration [22] featuring a 505 nm light-emitting diode (Roithner Lasertechnik B5B-433-B505, Farnell, Madrid, Spain) and a Hamamatsu S1337-66BR photodiode (Farnell, Spain).

NResearch three-way microvalves 161T031 (NResearch, West Caldwel, NJ, USA) were used for all fluid-handling operations, including multicommutation. These operations involved the preparation of stock solutions using automatic dilution, as well as the intake of samples, nitric acid cleaning solution, and water. Fluids were propelled using an ISMATEC peristaltic pump (ISM852A-115V60H, Cole-Parmer, Vernon Hills, IL, USA) connected to 1.2 mm Tygon (Ismatec Cole-Parmer GmbH, Nürtingen, Germany) and 0.8 mm i.d. polytetrafluorethylene (PTFE) (Tecnyflour, Terrassa, Spain) tubing.

The electronic control module was designed using EAGLE 9.3.1 software (Autodesk, San Francisco, CA, USA). The printed circuit of the module was designed using JLC software 5.4 (Shenzhen JLC Electronics Ltd., Shenzhen, China). The module was assembled and soldered by the authors. The main control element of the module is a programmable system on a chip, specifically the integrated circuit PSoC 5 CY8C5868AXI-LP035 (Infineon, Neubiberg, Germany). Figure 1 shows the general diagram of the experimental setup used to test the microfluidic devices.

### 2.1. Microfluidic Platform Fabrication Method

To achieve high precision and quality in 3D printing, the proper calibration of the printer is essential. Figure 2 shows the patterns used to ensure that the devices had the correct dimensions and to determine the optimal printing parameters, the thickness of the layers, and the exposure times for each of the resins [23].

Figure 3 shows a simplified layout of the printing process using UV photosensitive resin and SLA LCD technology. From left to right, in stage (1), the initial layer is cured. The build platform is dipped into the liquid resin pool at a distance equal to the thickness of the next layer. This layer of uncured resin, which is located between the previously cured layer and the bottom of the resin tank made of transparent fluorinated ethylene (FEP), is exposed to light from the curing lamp. The first few layers receive a longer exposure time to ensure good adhesion to the build plate, with care taken to ensure that the increasing weight of the printed device does not cause it to detach from the build plate. In some printers, the surface of the build plate is not smooth, presenting a rugged pattern that is transferred to the fixation layer. In stage (2), the build platform is raised, leaving the next layer of resin to be cured between the build plate and the FEP bottom of the resin tank. In stage (3), this next layer is irradiated and cured. The 3D structure arises from the succession of curing steps and joining of the layers.

To build microfluidic devices with channels of approximately 1 mm × 1 mm, two fabrication methods were employed. The first method consisted of the construction of a monolithic microfluidic platform in a single block. During the manufacturing process, the definition of the embedded microchannels via the selective irradiation of the layer in which they are integrated presents a challenge. Specifically, the microchannels remain filled with uncured resin. To prevent the photocuring of the resin trapped in the microchannels, both in the layer of the microchannels and in the additional layers deposited as a cover, a minimum light exposure time was applied [24]. Specifically, the exposure time for the two initial fixation layers was 40 s each, and the exposure time was 6 s for all the following layers. The thickness of each layer was 50 µm. For the second method, a multilamination approach was employed. Two separate blocks were printed and then glued together to form the microfluidic device. For this method, the layer thickness was also 50 µm, with exposure times of 50 s for the two fixation layers and 8 s for the remaining layers. The key considerations for the design of the layered devices are as follows:The full 3D design was generated using computer-aided design (CAD) software.The two blocks (halves) forming the device were defined. One block contains the open microchannels, which were printed to facilitate the removal of the uncured resin. The other block serves as a lid that seals the microchannnels.The individual blocks could have different thicknesses because they could have a different number of layers.Both blocks were joined via multilamination using a photocurable resin.

### 2.2. Removal of Uncured Resin

Once printed, the block with the structured microchannels was cleaned with isopropyl alcohol to remove the uncured resin that filled the microchannels. Then, the side of each block that contacted the build plate was polished to remove the rugged pattern transferred by the plate. Figure 4 shows the difference between a polished and a non-polished surface. The final assembly of the device was achieved by joining the two blocks with a thin layer of photocurable polymer, which was deposited on one of the faces (layers) to be assembled. The joining of the blocks was achieved by means of irradiation with no pressure applied, as the adhesive force of the uncured resin holds the two blocks together. Once aligned, the resin was photocured for 60 s at 60 W.

### 2.3. Assembly of Blocks

The layer of photocurable polymer used to join both halves of the 3D-printed microfluidic structure was deposited using a steel spatula (see Figure 5) in accordance with the following procedure:

The spatula is loaded by dipping it no more than 1 mm perpendicular to the surface of the resin.The spatula is withdrawn, and the excess resin is allowed to drain.The block with the largest surface area is chosen. This is generally the block that acts as a lid, closing the microchannels on the other block.The spatula is dragged at approximately 45 degrees to the surface where the resin is applied, taking care to spread it uniformly.Both blocks are aligned and joined without pressure to prevent the uncured resin from entering the microchannels.The microfluidic device is irradiated on both faces to attain a uniform and hermetic seal.

Figure 6 shows the pieces used to build a microfluidic module with approximately 0.8 mm × 0.8 mm channels. The module is designed to be used in a system like that reported by Guevara [2]. Placing the channels as shown avoids possible obstruction by resin draining due to gravity. The final dimension of the device is 54 mm × 44 mm.

## 3. Results and Discussion

All 3D prints were produced within a 54 mm × 44 mm area, with single-block prints having a thickness of 3 mm. The uncured resin inside the channels was removed by injecting isopropyl alcohol under positive pressure using a syringe. To prevent the photopolymerization of the resin within the channels during printing, it was necessary to reduce the exposure time for all layers to the minimum value recommended by the manufacturer. However, this reduction in exposure time led to insufficient adhesion between layers. The high viscosity of the uncured resin, combined with the pressure applied to flush it out, caused delamination, fractures, and leaks in microstructures with channels greater than 50 mm in length or with sharp turns designed to function as mixers. These defects are shown in Figure 7. The appearance of fractures and leaks during the channel cleaning process prevented the complete removal of uncured resin, rendering the module nonfunctional.

For the fabrication of devices using the multilamination-based approach, the dimensions of the channel designed were initially 2 mm × 2 mm. Then, the dimensions were decreased in 0.2 mm intervals until reaching channels 0.2 mm × 0.2 mm in size. The overall structure is composed of two blocks; one block incorporates the opened microfluidic channels, and the other acts as a cover block. In Figure 8, the different channels are shown without a cover on one of the blocks. Channels with dimensions equal to or greater than 0.4 mm presented good definition and repeatability. The printing of channels with dimensions of 0.2 mm was tested; however, blockages were generated in some sections, as shown in photograph (D) of Figure 8.

To evaluate the microfluidic structure produced using this method, several microchannels of identical dimensions were printed next to each other on the first block. Once both elements were joined, the resulting monolithic structure was cut, and the cross-section area of each channel was measured at various points.

Figure 9 shows photographs of the cross-sections of several printed and multilaminate monolithic structures obtained with different resins: (a) ELEGOO green resin; (b) ELEGOO transparent resin. In the amplified photograph shown in (c), it is evident that the joining of the two elements results in a single monolithic block, and the interface between them is not visible.

It was noted that the resin used to join the two elements penetrated the microchannel, causing a decrease in its height. The mean height reduction was 0.135 mm, with a standard deviation of 0.02 mm. The thickness of the sealing polymer layer after photocuring is less than 0.150 mm. This reduction must be considered during the design process to achieve the desired channel sizes.

The microfluidic structures developed using this methodology were ready for use after curing the resin layer used to join the two elements. The smallest channels constructed were 0.4 mm × 0.4 mm, regardless of the resin used, yielding leakage-free microfluidic platforms.

Figure 10 shows the design, distribution of masks, and number of layers used to manufacture the two blocks of the device shown in Figure 6. A layer thickness of 50 µm was employed, so the final design had a total thickness of 3.2 mm. The microfluidic platform comprised three fluidic inlets, two confluence points, two meander-based mixers, and one outlet.

For analytical applications using optical measurements, it is important to integrate detection cells into microfluidic platforms, allowing for the easy modification of the optical path. Given that the detection cell is formed by a section of the microfluidic channel, the fabrication methodology described above allows for the easy adjustment of the optical path length by varying the channel height. This modification not only reduces the dead volume of both the detection cell and the microfluidic system but also minimizes sample dilution, thereby enhancing measurement sensitivity. Figure 11 presents various fabricated channels that are 1 mm wide, with heights ranging from 0.25 mm to 2 mm.

Modifying the height/width ratio of the microchannels to increase the length of the optical path has other advantages. For instance, microchannels with ratios greater than 1 show greater resistance to deformation and are less susceptible to occlusion by the resin used as a sealant during the multilamination process.

Figure 12 shows the transmittance spectrum obtained with 3 mm thick plates using the four ELEGOO resins (polished clear, clear, green, red, and black) and the translucent ANYCUBIC resin. The 505 nm line shown in Figure 12 corresponds to the wavelength of maximum absorbance of the copper complex obtained with the colorimetric agent used in the experiment to validate the fluidic module. The transmittance was less than 30% for all the studied resins. This fact is related to the surface roughness of the first layer printed in contact with the build platform.

Additional experiments conducted with the transparent ELEGOO resin, after polishing the rough surfaces, revealed that the transmittance increased by more than double, reaching values exceeding 47% for wavelengths above 425 nm and surpassing 60% for wavelengths greater than 540 nm. As noted, the polishing step is critical for enhancing the measured signal, thereby improving the sensitivity and detection limit of the analytical microsystem.

To validate the usefulness of the developed 3D-printed microfluidics using the described multilamination process, the microfluidic platform (as described in Section 2 and Figure 6 and Figure 10) was designed with a cross-sectional channel of 0.8 mm × 0.8 mm for the colorimetric determination of copper using a method previously described by Guevara [2].

This microfluidic platform was incorporated into a multicommutation flow system microanalyzer, which performs automatic sampling and autocalibration procedures similar to those described in [2].

Working under constant flow conditions, the sample volume injected depends on the actuation time of the three-way valves, which select between the sample/standard solution and the carrier solution through commutation. To ensure robust microfluidic platforms that remain unaffected by slight changes in experimental conditions, it was necessary to select a sample injection time that allowed the analytical signal to stabilize at its steady state. Figure 13 shows the results of the experiment performed at a fixed flow rate of 1.4 mL/min in each channel (Q1, Q2, and Q3) to determine the optimal injection time. The sample injection time selected was 160 s.

Finally, the multicommutation approach enables the automatic execution of an autocalibration procedure through the “in situ” generation of different standard solutions from a single concentrated stock solution.

Analytical features derived from the calibration plot obtained in the experimental conditions fixed after the optimization process show a linear response in the range of 3 to 24 ppm of copper (II), with a correlation coefficient r2 > 0.999. Figure 14 shows the analytical signals obtained during the autocalibration cycle featuring triplicate measurements.

To assess the reproducibility of the analytical signals provided by different microfluidic platforms operating under identical conditions, four platforms were tested over a twelve-week period. All of them yielded similar results. The dimensions, polymeric resin, coloring, and structural integrity (no breakage or leakage) were maintained in all cases. The analytical results obtained with the devices fabricated using the 3D printing technology were comparable to those provided by microfluidic platforms made of COC using the multilamination approach. The main difference was the cost; the microfluidic platforms described here were more affordable in terms of both materials and infrastructure requirements.

## 4. Conclusions

The methodology described for the fabrication of microfluidic systems, combining the advantages of resin-based 3D printing (using commercial printers) and multilamination procedures, enables the simple and rapid construction of microfluidic platforms with more complex and extended microchannel pathways. The proposed method, based on joining multiple pre-printed blocks, yielded better results than traditional single-piece printing of the microfluidic module. This is primarily due to the challenge of evacuating uncured resin trapped in the microchannels after printing.

All structures and modules were fabricated using a commercial Epax printer (model X1) and a Sunlu UV curing chamber. With an average fluidic module volume (see Figure 6) of 5 mL, the number of printable pieces per 500 mL (commercial resin packaging) highlights the cost-effectiveness of the modules, aligning with the goal of producing low-cost microfluidic devices. During block bonding, the channel height decreased by approximately 0.135 mm (with a standard deviation of 0.02 mm) due to the use of the resin as an adhesive. This phenomenon must be accounted for during the design stage to ensure correct channel dimensions.

Based on the results obtained, it is recommended that the uncured resin used to bond the blocks be applied to the face that forms the ceiling of the channels in the microfluidic module. For optical applications, it is essential to polish the surface of the blocks that were in contact with the printer’s build plate. This post-processing step ensures a significant increase in the optical transmittance of the entire module.

Throughout the fabrication process, only CAD software (Fusion 360) and the 3D printer’s slicing software (Chitubox 1.9.4) were required. Block bonding was performed at room temperature without applying pressure, resulting in seamless cohesion between the parts. In addition, no visible interface was observed where the blocks were joined (Figure 9).

In this study, a multicommutation flow system microanalyzer for copper detection incorporating the fabricated microfluidic platforms was evaluated and optimized. Its operational performance and analytical features were identical to those of platforms constructed with COC. The achievement of flat surfaces with enhanced optical transmittance obtained via polishing, along with the quality of layer adhesion during the post-processing stage, opens the possibility of combining materials and technologies for future analytical applications.

## Figures and Tables

**Figure 1 sensors-25-00694-f001:**
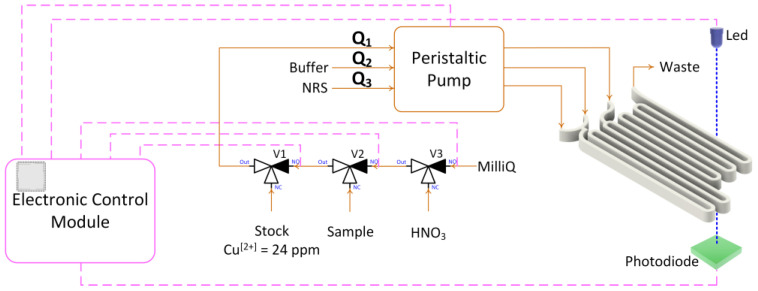
Copper microanalyzer: peristaltic pump; three-way valves (V_1_, V_2_, and V_3_); microfluidic platform; detector (LED 505 nm and photodiode); electronic control module and waste. **---** indicates electronic connections between different modules.

**Figure 2 sensors-25-00694-f002:**
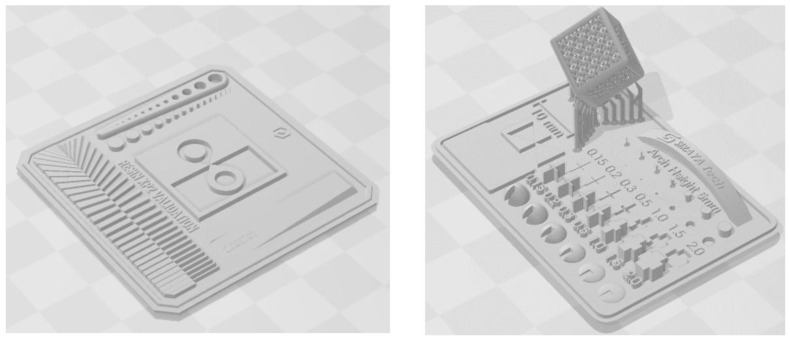
Patterns used for lamination parameter calibration.

**Figure 3 sensors-25-00694-f003:**
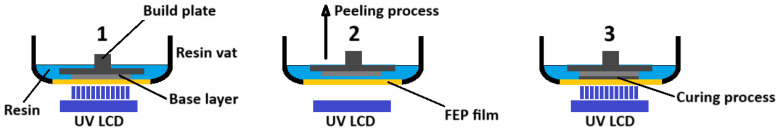
Simplified diagram of the 3D printing process using SLA LCD.

**Figure 4 sensors-25-00694-f004:**
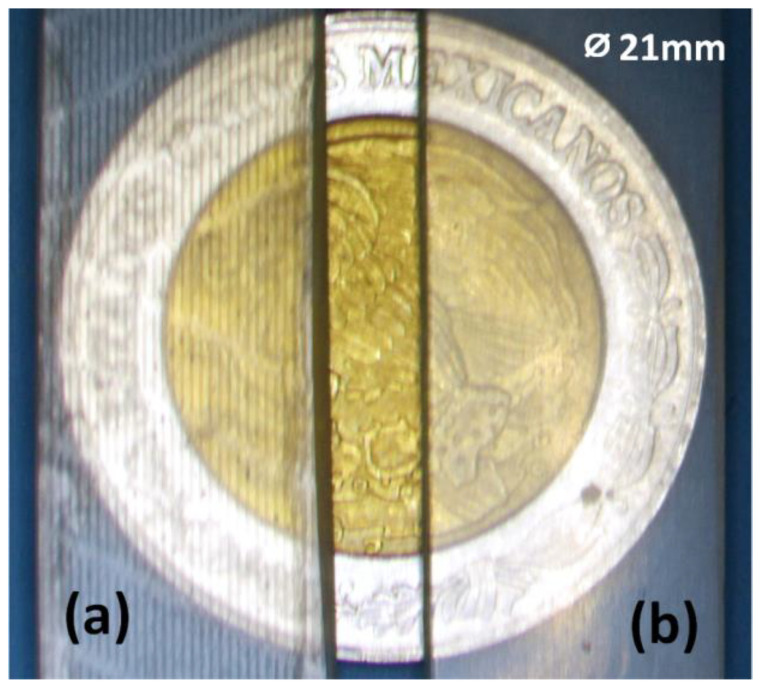
Final surface of the microfluidic platform before (**a**) and after (**b**) polishing.

**Figure 5 sensors-25-00694-f005:**
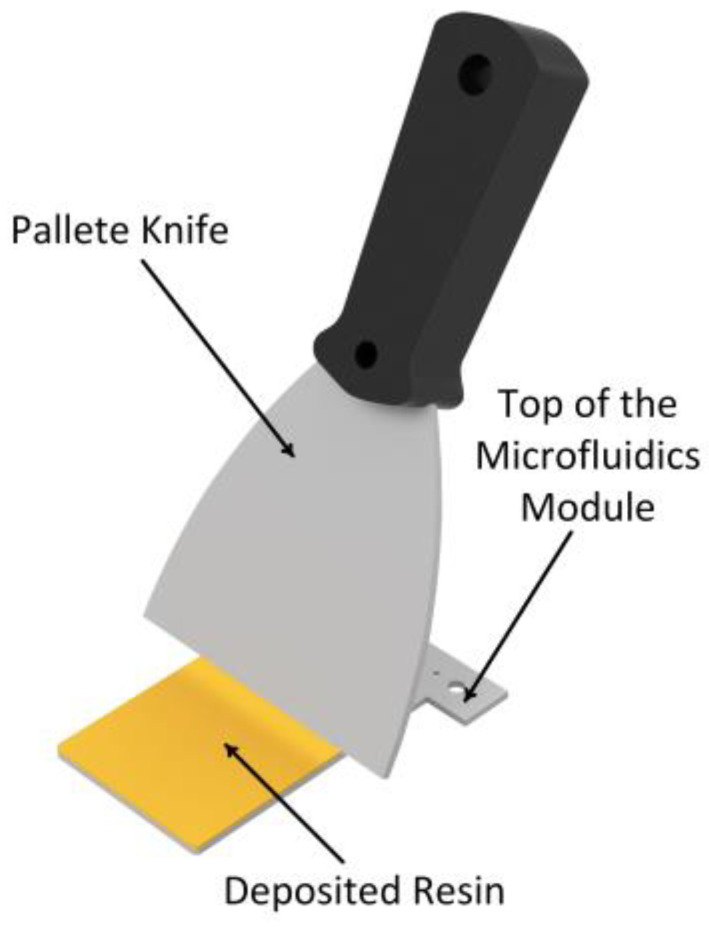
Image showing how the resin is applied to block 1 for bonding to block 2.

**Figure 6 sensors-25-00694-f006:**
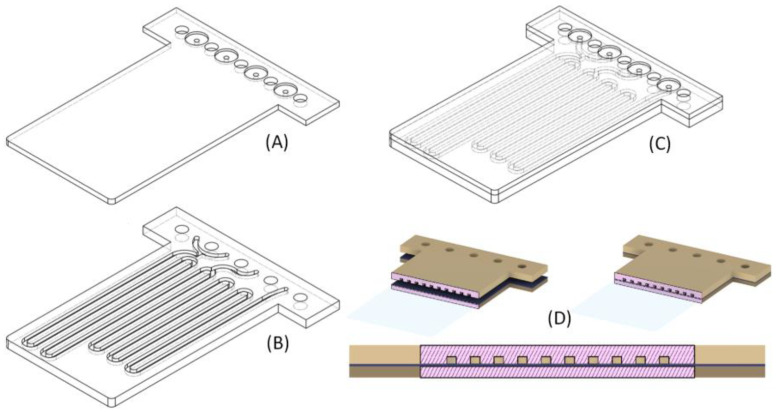
Example of a multilayered microfluidic design. (**A**) The block to which uncured resin is applied as an adhesive. (**B**) The block that incorporates microchannels. (**C**) The bonding of the two blocks is achieved by placing block (**A**) on top of block (**B**). (**D**) A cross-section of both blocks is shown during the alignment and bonding process.

**Figure 7 sensors-25-00694-f007:**
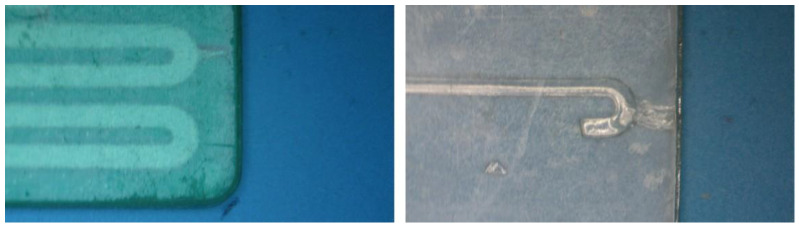
Layer separation and leakage in the microfluidic structure when pressure was applied to remove the uncured resin from the microchannels.

**Figure 8 sensors-25-00694-f008:**
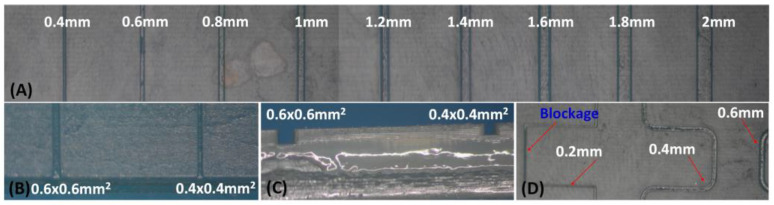
(**A**) Zenithal view of the substrate incorporating straight open channels ranging from 0.4 mm^2^ to 2 mm^2^. (**B**) Amplified zenithal view of the channels 0.6 × 0.6 mm^2^ and 0.4 × 0.4 mm^2^ in size. (**C**) Amplified cross view of the channels ranging from 0.6 × 0.6 mm^2^ to 0.4 × 0.4 mm^2^ in size. (**D**) Zenithal view of the substrate incorporating curved open channels ranging from 0.2 mm^2^ to 0.6 mm^2^.

**Figure 9 sensors-25-00694-f009:**
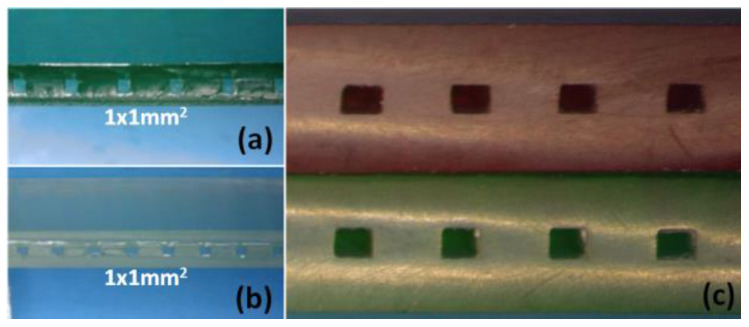
Channel cross-section view of the monolithic microfluidic structures. (**a**) 1 mm × 1 mm straight microchannel (ELEGOO green photocurable resin). (**b**) 1 mm × 1 mm straight microchannel (ELEGOO clear photocurable resin). (**c**) Amplified view of the monolithic microfluidic block after the multilamination process.

**Figure 10 sensors-25-00694-f010:**
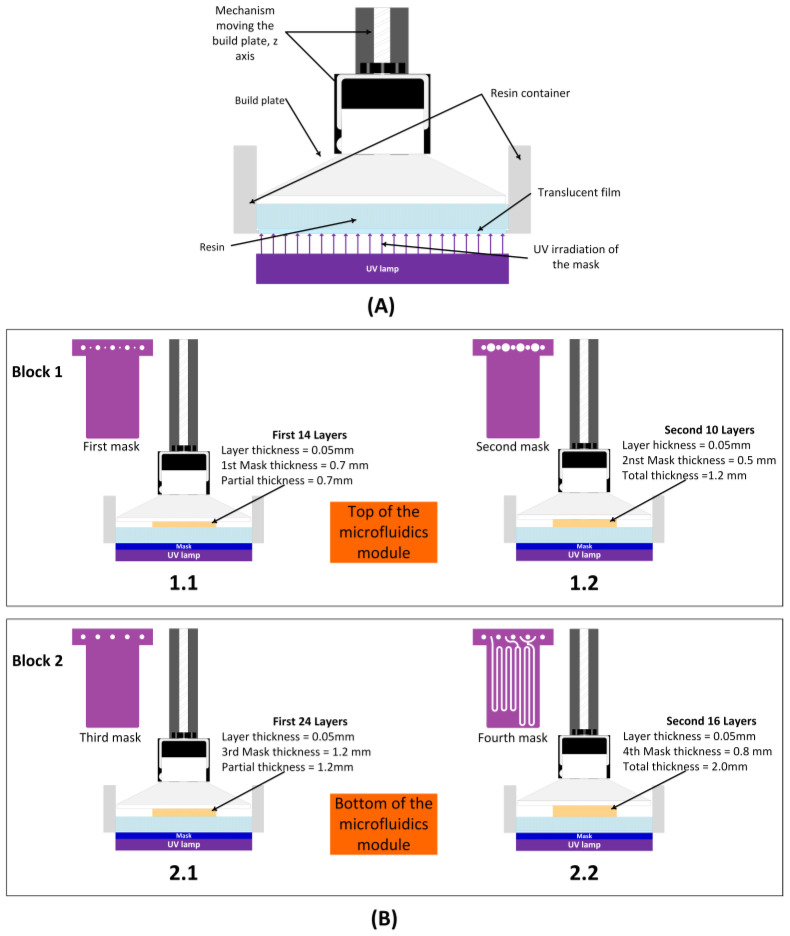
Procedure followed for the 3D printing of the device shown in Figure 6. (**A**) Experimental setup. (**B**) Design, distribution of masks, and number of layers used for the fabrication of blocks 1 and 2.

**Figure 11 sensors-25-00694-f011:**
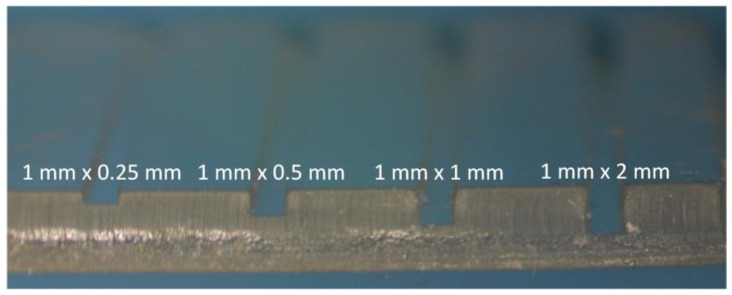
Three-dimensional printed layer with open channels of different height/width ratios and optical path lengths.

**Figure 12 sensors-25-00694-f012:**
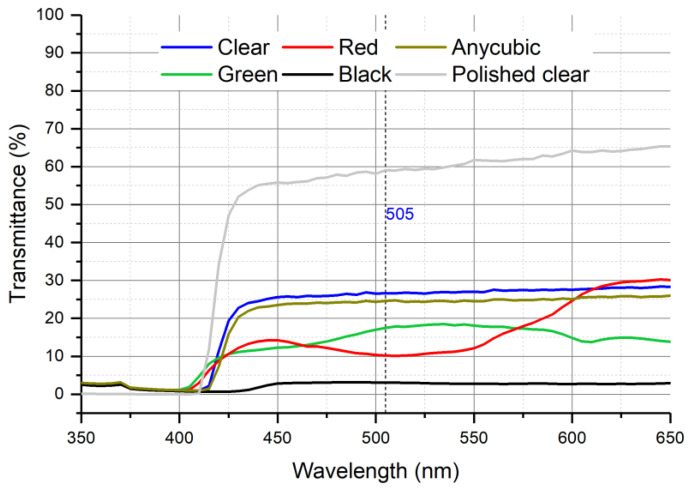
Transmittance spectrum using different resins.

**Figure 13 sensors-25-00694-f013:**
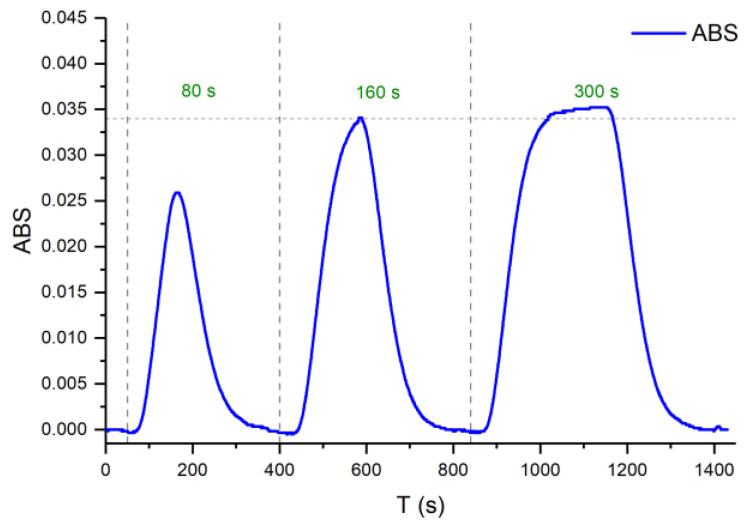
Analytical signal used to determine the injection time with a flow rate of 1.4 mL/min.

**Figure 14 sensors-25-00694-f014:**
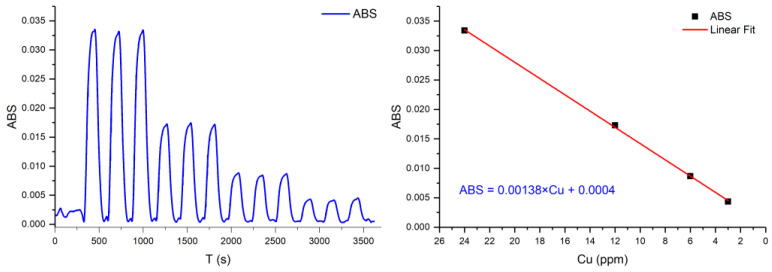
Analytical response of a calibration run and the resulting calibration curve.

## Data Availability

The original contributions presented in this study are included in the article. Further inquiries can be directed to the corresponding author.

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
