# Peer review of "Multilayered Manufacturing Method for Microfluidic Systems Using Low-Cost, Resin-Based Three-Dimensional Printing"

_sensors, 2025, doi:10.3390/s25030694_

Round 1

Reviewer 1 Report

Comments and Suggestions for Authors

1.The numbers of references suggest to be increased.

2.In your abstract section, the performance of your microfluidic systems suggest to be supplemented in your abstract to reflect the importance of your paper.

3.The authors mention in the last sentence of your abstract this study highlights the potential of 3D printing as a flexible, efficient, and cost-effective alternative for the fabrication of customized microfluidic devices, if the data of flexible, efficient, and cost-effective microfluidic systems were provided, the statement will be more convincing for readers.

4.In your section of Introduction, the format of writing should be modified, and a sentence cannot become a paragraph, for example paragraph 2 and 3, 9, 10, 11, 14 and so on, one sentence is as a paragraph.

5. In the introduction section of paragraph 2 to 7, the authors summarize the methods for fabrication microfluidic systems, such as CNC, microembossing and microinjection, whats those methods advantages and disadvantages suggested to add in your paper. And whats the relationship with 3D printing method for fabrication microfluidic systems and whats your advantages?

5. The introduction section should be rewrote to state your necessity of your research.

6. In your last paragraph of your introduction, the materials and methods, some valuable research results and the promising applications should be added in this paragraph.

7.The simplify and condense your experimental results is necessary in your conclusion section and this section is suggested modification.

8.The language suggested to be polish by the native speaker, some sentence in your paper is elusive.

Comments on the Quality of English Language

The language suggested to be polish by the native speaker, some sentence in your paper is elusive.

Author Response

Comments 1: The numbers of references suggest to be increased.

Response 1: Thank you for pointing this out. We agree with this comment. Therefore, we have… The number of references was increased from 18 to 28, as reflected on pages 13 and 14, lines 470–528.

Comments 2: In your abstract section, the performance of your microfluidic systems suggest to be supplemented in your abstract to reflect the importance of your paper.

Response 2: Agree. We have, accordingly, modified…to emphasize this point. The abstract was rewritten to emphasize the importance of the microfluidic system’s performance (page 1, first paragraph, lines 12–25).

Comments 3: The authors mention in the last sentence of your abstract this study highlights the potential of 3D printing as a flexible, efficient, and cost-effective alternative for the fabrication of customized microfluidic devices, if the data of flexible, efficient, and cost-effective microfluidic systems were provided, the statement will be more convincing for readers.

Response 3: Thank you for pointing this out. We agree with this comment. Therefore, we have…The abstract was also revised to clearly explain the system’s flexibility, efficiency, and cost-effectiveness (page 1, first paragraph, lines 12–25).

Comments 4: In your section of Introduction, the format of writing should be modified, and a sentence cannot become a paragraph, for example paragraph 2 and 3, 9, 10, 11, 14 and so on, one sentence is as a paragraph.

Response 4: Thank you for pointing this out. We agree with this comment. Therefore, we have…One-sentence paragraphs were removed (e.g., page 1, paragraph 2, lines 37 and 40). Since this issue appeared multiple times, similar corrections were applied throughout the document.

Comments 5: In the introduction section of paragraph 2 to 7, the authors summarize the methods for fabrication microfluidic systems, such as CNC, microembossing and microinjection, what’s those methods advantages and disadvantages suggested to add in your paper. And what’s the relationship with 3D printing method for fabrication microfluidic systems and what’s your advantages?

Response 5: Agree. We have, accordingly, modified…to emphasize this point. Several paragraphs were revised to better explain the advantages and disadvantages of various fabrication methods and their relevance to the method used in this study (page 1, paragraphs 3–11, lines 41–107).

Comments 5: The introduction section should be rewrote to state your necessity of your research.

Response 5: Agree. We have, accordingly, modified…to emphasize this point. The introduction was modified to highlight the significance of the study (pages 1–3, paragraphs 2–16, introduction section, lines 37–134).

Comments 6: In your last paragraph of your introduction, the materials and methods, some valuable research results and the promising applications should be added in this paragraph.

Response 6: we don't Agree. We have, accordingly, revised ….to emphasize this point.

We consider that the methods and materials, the results and future applications are contained in the relevant sections. 

Comments 7: The simplify and condense your experimental results is necessary in your conclusion section and this section is suggested modification.

Response 7: Agree. We have, accordingly, modified…to emphasize this point. The conclusions section was rewritten to simplify and condense the key findings (page 12, paragraphs 1–5, lines 414–446).

Comments 8: The language suggested to be polish by the native speaker, some sentence in your paper is elusive.

Response 8: Agree. We have, accordingly, modified…to emphasize this point. After making all the corrections, the manuscript underwent professional English editing using MDPI’s editing service. The revised document incorporates all recommendations provided by their editor.

Reviewer 2 Report

Comments and Suggestions for Authors

1. a secondary heading like 2.1 is used in line 170, but I don't find a serial number for 2.2 later in the text.

2. heading 3 in line 261 uses RESULTS, but I think this part is the analysis process and suggest changing it to ANALYSIS or something else.

3. In the paragraph from line 262 to 271, you say that some leakage will occur, but nothing specific?Or is there something else you want to clarify in this paragraph.

4. The picture in line 285 does not contain 0.2*0.2mm channels, but you said in the paragraph above that there is such an experiment, although he is not well defined and repeatable, I suggest that you can add a picture of such conditions, so that the reader will also know what often happens when it is less than 0.4*0.4mm.

5. The descriptions in lines 346 to 353 do not correspond to the following diagrams, I see that the green color is maximal at 540nm, and the polished ones also achieve more than 60% transmittance at 540nm.

Author Response

Comments 1: a secondary heading like 2.1 is used in line 170, but I don't find a serial number for 2.2 later in the text.

Response 1: Thank you for pointing this out. We agree with this comment. Therefore, we have… Two subsections were omitted, and the remaining two were added on page 6, lines 229 and 243.

Comments 2: heading 3 in line 261 uses RESULTS, but I think this part is the analysis process and suggest changing it to ANALYSIS or something else.

Response 2: Agree. We have, accordingly, modified…..to emphasize this point. The section title was changed to “Results and Discussion” page 7 , line 271.

Comments 3: In the paragraph from line 262 to 271, you say that some leakage will occur, but nothing specific? Or is there something else you want to clarify in this paragraph.

Response 3: Agree. We have, accordingly, modified….to emphasize this point. The writing was improved on pages 7 and 8, first paragraph of the results and discussion section
, lines 271–282.

Comments 4: The picture in line 285 does not contain 0.2*0.2mm channels, but you said in the paragraph above that there is such an experiment, although he is not well defined and repeatable, I suggest that you can add a picture of such conditions, so that the reader will also know what often happens when it is less than 0.4*0.4mm.

Response 4: Agree. We have, accordingly, modified….to emphasize this point. A photograph was included to show a channel with dimensions of 0.2 mm × 0.2 mm. The photograph was labeled as “D.” Additionally, the text was clarified to better explain when cracks and leaks occur (page 8, first paragraph of the results and discussion section, lines 293–295 and 300–302).

Comments 5: The descriptions in lines 346 to 353 do not correspond to the following diagrams, I see that the green color is maximal at 540nm, and the polished ones also achieve more than 60% transmittance at 540nm.

Response 5: Agree. We have, accordingly, modified….to emphasize this point. The indicated sentence was corrected to clarify that 505 nm corresponds to the wavelength of maximum absorbance for the colorimetric agent used in the experiment to validate the 3D-printed module. The frequency sweep data demonstrated that polished resin exhibits good transmittance at this wavelength (page 10, paragraph 10, lines 356–362). A reference to the graph in Figure 12 was also added (page 10, line 367).

Round 2

Reviewer 2 Report

Comments and Suggestions for Authors

In the paragraph above Figure 12, you said "showed a transmittance of greater than 60% for wavelengths above 425 nm." However, as can be seen from the graph, the transmittance only reaches about 50% at wavelengths above 425 nm. This question must be clarified.

Author Response

Comments 1: In the paragraph above Figure 12, you said "showed a transmittance of greater than 60% for wavelengths above 425 nm." However, as can be seen from the graph, the transmittance only reaches about 50% at wavelengths above 425 nm. This question must be clarified.

Response 1: Thank you for pointing this out. We agree with this comment. Therefore, we have… The indicated paragraph was modified, improving the description of the results in relation to the graph, as shown in paragraph 11, page 10, lines 363 to 366.

"Additional experiments conducted with the transparent ELEGOO resin, after polishing the rough surfaces, revealed that the transmittance increased by more than double, reaching values exceeding 47% for wavelengths above 425 nm and surpassing 60% for wavelengths greater than 540 nm. As noted, the polishing step is critical for enhancing the measured signal, thereby improving the sensitivity and detection limit of the analytical microsystem."